# Initial Experience of Robot-Assisted Transabdominal Preperitoneal (TAPP) Inguinal Hernia Repair by a Single Surgeon in South Korea

**DOI:** 10.3390/medicina59030582

**Published:** 2023-03-15

**Authors:** Yun Suk Choi, Kyeong Deok Kim, Moon Suk Choi, Yoon Seok Heo, Jin Wook Yi, Yun-Mee Choe

**Affiliations:** Department of Surgery, Inha University Hospital & College of Medicine, Incheon 22332, Republic of Korea

**Keywords:** inguinal hernia, robotic surgery, hernia surgery

## Abstract

*Background and Objectives*: Inguinal hernia is a common surgical disease. Traditional open herniorrhaphy has been replaced by laparoscopic herniorrhaphy. Nowadays, many attempts at robotic herniorrhaphy have been reported in western countries, but there have been no reports in South Korea. The purpose of this study is to report our initial experience with robotic inguinal hernia surgery, compared to laparoscopic inguinal hernia surgery. *Materials and Methods*: We analyzed the clinical data from 100 patients who received inguinal hernia surgery in our hospital from November 2020 to June 2022. Fifty patients underwent laparoscopic surgery, and 50 patients underwent robotic surgery using the da Vinci Xi system. All hernia surgeries were performed by a single surgeon using the transabdominal preperitoneal (TAPP) method. *Results*: The mean operation time and hospital stay were not statistically different. On the first postoperative day, the visual analog scale (VAS) pain score was significantly lower in the robotic surgery group (2.9 ± 0.5 versus 2.5 ± 0.7, *p* = 0.015). Cumulative sum analysis revealed an approximately 12-case learning curve for robotic-assisted TAPP hernia surgery. *Conclusions*: Robotic-assisted TAPP inguinal hernia surgery is technically acceptable to surgeons who have performed laparoscopic inguinal hernia surgery, and the learning curve is relatively short. It is thought to be a good step toward learning other robot-assisted operations.

## 1. Introduction

Inguinal hernias have a lifetime incidence of 25% among men and 3% among women [1,2]. In the United States, 700,000 to 800,000 hernia repairs are performed annually. In South Korea, about 34,000 such annual repairs are performed out of about 50,000 patients treated for inguinal hernias. [1,2,3,4,5] Inguinal hernia causes discomfort to the patient as it progresses, and bowel strangulation may occur in severe cases. Hernia is a problem caused by a structural defect in the human body, and surgery to correct it is performed as the gold standard of treatment.

Open hernia surgery was the mainstay of hernia repair until the late 1990s. The tension-free and Lichtenstein methods of open hernia repair were introduced to reduce postoperative pain and recurrence. [1] After the development of endoscopic surgery, laparoscopic inguinal hernia surgery was first reported in 1991 by Ger et al. [6] The rate of laparoscopic hernia surgery has been gradually increasing recently, and it is considered the primary operation choice by some surgeons. In South Korea, more than 40% of inguinal hernia operations are performed laparoscopically. [4,5] The advantages of laparoscopic hernia repair include the following: it is associated with less pain, smaller wounds, lower recurrence rates, earlier returns to work, school, and activities of daily living, and lower complication rates. Also, bilateral hernia surgery is possible with the same incision with a laparoscopic approach [7,8].

Several studies have demonstrated the advantages of robotic surgery, such as 3-dimensional visualization, the elimination of tremor, EndoWrist’s precise and free movement, and improved surgeon comfort and performance. [7,8,9,10] Given these advantages, robotic surgery has been applied to a variety of surgical contexts, and this has further confirmed its safety and efficacy. [11,12,13,14] The first report of robot-assisted hernia surgery was of a repair performed together with urologic procedures in 2014, and Dominguez et al. reported the first robotic hernia surgery results. [1,15] Since then, robotic hernia surgery has been reported to reduce postoperative pain and facilitate ergonomic optimization for surgeons, and it has been attempted and adopted by surgeons worldwide [1,15,16,17].

Our institution has performed the most robotic inguinal hernia surgeries in South Korea. This study aimed to evaluate the safety and effectiveness of robotic inguinal hernia surgery compared with laparoscopic inguinal hernia surgery and the learning curve for robotic inguinal hernia surgery.

## 2. Materials and Methods

### 2.1. Study Design

We retrospectively analyzed the electronic medical data of patients who underwent laparoscopic or robot-assisted inguinal hernia surgery at our hospital between November 2020 and June 2022. The surgical candidates for robotic inguinal hernia surgery were the same as those for laparoscopic hernia surgery. The choice was left to the patients after they were provided with detailed information about both options. Robotic inguinal hernia surgery was chosen when the patient agreed to the procedure despite its higher expense compared to laparoscopic surgery. We reviewed clinical data, including age, gender, body mass index (BMI), preoperative clinical diagnosis, preoperative morbidity, preoperative hernia surgery history, hernia type, operation time, postoperative pain, hospital stay, postoperative complications, and cost of surgery. All surgical videos were recorded, and we reviewed all of these videos. We captured video clips of important moments and determined actual operation times using the surgical videos. In cases of combined other operations, operation time and cost of surgery data were only evaluated for the hernia surgery. The cost of surgery variable only took into account the fees paid for surgery and excluded extra expenses, such as hospitalization fees and others.

### 2.2. Patients

From November 2020 through June 2022, 100 cases of robotic and laparoscopic inguinal hernia surgery were performed by a single surgeon (YS Choi) at the authors’ hospital. All patients underwent inguinal hernia surgery using the transabdominal preperitoneal (TAPP) approach. The analysis considered two groups: the laparoscopic inguinal hernia repair (LIHR) group and the robotic inguinal hernia repair (RIHR) group. There were 50 patients in each group.

The surgeon for all of the cases is a gastrointestinal surgeon who has performed more than 100 cases of LIHR using totally extraperitoneal (TEP) and TAPP approaches. After the introduction of robotic-assisted TAPP hernia surgery at our hospital. This was the main technique of choice except when there were contraindications such as previous abdominal surgery history, previous laparoscopic hernia surgery history, or a poor condition for general anesthesia.

### 2.3. Statistics and Ethical Considerations

All statistical analyses were performed using SPSS Statistics for Windows, version 28 (IBM Corp., Armonk, NY, USA). Continuous variables are presented as means ± standard deviations. Unpaired t-tests were used to compare means. The chi-square test, or Fisher’s exact test, was applied to the cross-table analysis according to the sample size.

The ethics of this study were approved by the Institutional Review Board of Inha University Hospital (IRB number: INHAUH 2022-11-024).

### 2.4. Cumulative Sum (CUSUM) Analysis

The operation times of robotic-assisted TAPP operations were analyzed from the start of 50 cases. All surgical videos were reviewed, and the console times of all robotic procedures were determined. In the case of bilateral hernias, we considered the need for surgery only on the more severe side. The learning curve was assessed based on the console times. Inflection points were based on each set of three or more consecutive negative values. Based on these inflection points, the learning curve was divided into pre-adapted and post-adapted phases.

### 2.5. Surgical Procedure for Robotic-Assisted TAPP Inguinal Hernia Repair

All operations were performed via the TAPP approach. Under general anesthesia, patients were placed in the supine position. Three trocars were inserted (Figure 1). The trocar locations were as follows: a camera port was inserted through the umbilicus, and two operating ports were inserted at both lateral aspects of the rectus abdominis muscle. A 12-mm trocar was used to insert mesh or gauze through the umbilicus. We tried to hide a 12-mm trocar wound using the umbilicus. The same locations were used for the robotic and laparoscopic groups. A distance of about 10 cm was maintained between the trocars.

In the RIHR group, the camera was first inserted, and then targeting was performed on the hernia site. In bilateral hernia repairs, targeting was performed in the middle of both hernias. To prevent collision of the robot arms, we spaced the robotic arms as far apart as possible, and we performed robotic machine docking (Figure 2). We use three types of robotic EndoWrist instruments: prograsp forceps for tissue grasping, monopolar curved scissors for dissecting the peritoneum, and mega needle holders for dissecting the hernia sac and suturing the peritoneum.

The steps of RIHR are shown in Figure 3. The peritoneum was dissected using monopolar curved scissors (Figure 3A). At this time, we were careful to avoid damage to the inferior epigastric vessels, and we started peritoneal dissection adjacent to the vessels. We performed a sufficiently wide area dissection that included the hernia site and inserted a 15.7 X 10.3 cm large size mesh (3DMax™ Light Mesh,1 Becton drive Franklin Lakes, NJ 07417, USA) without wrinkles. In the case where there were adhesions of the omentum or bowel around the hernia site during surgery, we performed adhesiolysis before peritoneal dissection.

After peritoneal dissection, we identified the hernia sac and dissected the hernia sac from the vas deferens and testicular vessels. A large-sized mesh was applied to a large enough area to include the entire hernia site, and we fixed the mesh onto the pubic bone using a tacker. To prevent injury by tacker slippage during fixation, we immobilized the tacker with a robot arm before firing the tacker (Figure 3B).

For indirect hernia cases, we performed hernia sac inversion after hernia sac dissection, and we sutured the hernia sac together during reperitonization. For direct hernia cases, we fixed the hernia sac to the pubic bone using a tacker to flatten the hernia site (Figure 3C), and we applied a large-size mesh to the pubic bone with a tacker. After reperitonization, we confirmed that there were no defects in the peritoneum without exposure to mesh, and we completed the operation (Figure 3D).

## 3. Results

Table 1 summarizes the clinical characteristics of the 50 patients who underwent LIHR and the 50 patients who underwent RIHR. The mean age of the LIHR group was significantly higher than that of the RHIR group (64.40 ± 14.83 versus 54.40 ±13.97 years; *p* = 0.001). There were no differences in gender ratio or mean BMI between the two groups. The mean American Society of Anesthesiology (ASA) score indicating preoperative condition was significantly higher in the LIHR group. (2.32 ± 0.55 versus 2.02 ± 0.38; *p* = 0.002). The proportion of patients with ASA III (indicating patients with severe systemic disease) was significantly higher in the LIHR group (18/50 (36.0%) versus 4/50 (8.0%); *p* = 0.003). The LIHR group had significantly more patients with hypertension (HTN) than the RIHR group, but there were no significant differences in terms of other underlying diseases and lifestyle factors between the two groups.

Table 2 summarizes the operational details of the two groups. There was no significant difference in mean operation time between the LIHR and RIHR groups (31.52 ± 10.31 versus 30.22 ± 11.87 min; *p* = 0.56). The mean cost of surgery was significantly higher in the RIHR group (209.61 ± 27.52 US dollars versus 3814.75 ± 172.97 US dollars; *p *< 0.001). The RIHR group’s cost of surgery includes the cost of several consumables, including a $400 laparoscopic tacker. All operations were completed according to the existing planned surgical method, without open or laparoscopic conversion. There was no significant difference in hernia type between the 2 groups. Indirect hernias were the most common, followed by direct hernias. There were 2 cases of combined hernias in the RIHR group and 1 case in the LIHR group. There was 1 case of femoral hernia in the RIHR group and 1 case of spigelian hernia in the LIHR group. Right-sided inguinal hernias were more common than left-sided inguinal hernias in both groups. There were 5 cases of bilateral inguinal hernias in the LIHR group and 3 cases in the RIHR group, respectively.

The postoperative outcomes are summarized in Table 3. There were no significant intergroup differences in hospital stays, readmission rates within 30 days, or hernia recurrence rates. Postoperative pain was evaluated using a visual analog scale (VAS). There was no difference in VAS pain scores on operation day. On postoperative day 1, the VAS pain score was statistically significantly lower in the RIHR group (2.86 ± 0.54 versus 2.54 ± 0.73; *p* = 0.015). Postoperative seroma and hematoma formation occurred more frequently in the LIHR group, but there was no significant difference. Urinary retention was also more common in the LIHR group, but again, there was no significant difference between the groups (5/50 (10.0%) versus 3/50 (6.0%); *p* = 0.461).

The console time of robotic inguinal hernia surgery was analyzed using CUSUM analysis (Figure 4). The inflection point was measured at approximately 12 cases. After the inflection point, it was confirmed that the CUSUM score decreased continuously. We compared the operation times between the pre-adapted phase and the post-adapted phase based on the inflection point. The mean operation time was shorter in the post-adaptation phase than in the pre-adaptation phase, but this difference was not statistically significant (35.50 ± 17.01 versus 28.55 ± 9.41 min; *p* = 0.077).

## 4. Discussion

Many studies have evaluated the advantages and disadvantages of robotic inguinal hernia surgery. Robotic inguinal hernia surgery is more expensive and takes longer than conventional laparoscopic inguinal hernia surgery, and it is not conducive to operator ergonomics. [18] Several articles have reported relatively long operation times but low postoperative complication rates and pain levels associated with robotic inguinal surgery. [16,17,19,20] The Da Vinci Xi system dramatically reduced robot docking time compared with the previous Si system. In our study, the actual docking time was about 2 min. Considering that there was little difference between the laparoscopic operation time and the robot console time (31.5 ± 10.3 versus 30.2 ± 11.9 min; *p *= 0.56), the mean total operation time was not different between the two groups. After passing the learning curve, the mean operation time decreased to 28 min, and there was no statistically significant difference in operation time between the pre-adaptation phase and the post-adaptation phase. (31.52 ± 10.31 versus 28.55 ± 9.41 min; *p* = 0.169). The long operation times mentioned in previous publications did not apply to our study. In our study, the mean pain score on the first postoperative day was lower in the RIHR group. Postoperative complication rates were lower in the RIHR group, but there was no statistical difference.

The learning curve for robotic inguinal hernia surgery was about 12 cases in our study, which was similar to what has been reported elsewhere. [20] However, longer learning curves have also been reported. [21] Laparoscopic inguinal hernia surgery is a frequently performed operation. [1,2,3,4,5] Robotic inguinal hernia surgery has favorable characteristics, such as short operation times, a short learning curve, a relatively fixed view, and minimal equipment requirements. Considering this, robotic inguinal hernia surgery is thought to be a good option for a first procedure for surgeons learning robotic surgery.

The 3-dimensional augmented view of the robotic surgery system is helpful for protecting the vas deferens and testicular vessels. The EndoWrist movement of the robotic arm facilitates efficient removal of the hernia sac without damaging these structures. Old and severe hernias are associated with difficult hernia sac dissections due to severe adhesions. The free movement and strong force of the robotic arm make this easier, and these features are very useful for the excision of huge cord lipomas. The robotic surgery system is helpful for reperitonization after mesh application. The free movement of the robotic arm facilitates reverse suturing and complete reperitonization of the injured peritoneum during hernia sac dissection without mesh exposure. The conventional laparoscopic TAPP approach is inconvenient for human ergonomics due to the surgeon’s posture being very uncomfortable; however, this ergonomic inconvenience has been improved, and the operator is now able to perform the operation in a more comfortable position. We plan to conduct a study investigating surgical ergonomics in this context using intraoperative surgeon electromyography (EMG) in the future [22,23,24,25].

The central camera port was inserted transumbilically to minimize scarring, and the two ports for the remaining robotic arm were inserted into the lateral aspects of the rectus abdominis muscle. Using these port locations, it is easy to operate on incidentally discovered contralateral hernias. In addition, surgeries such as cholecystectomy can be performed only by changing the direction of the robot docking without changing the port site. We performed robotic-assisted TAPP inguinal repair on 2 patients with morbid obesity (BMI ≥35). In such cases, it is difficult to secure the operative field because of the severe visceral obesity, so an additional assist port is used. The intuitive guideline recommended that an assist port be inserted at the level of the epigastric area between the two robot arms. In this situation, an assistant must be placed between the robot arms, which can lead to frequent extracorporeal fighting between the robot arm and the assistant. In our experience, insertion of the assistant on the lateral side of the arm opposite the hernia site reduces this extracorporeal fighting. The para-umbilical camera port is considered to move toward the hernia site in morbidly obese patients. This can help further centralize the target anatomy and avoid a thick pannus and preperitoneal fat layer over the median umbilical ligament in obese patients. This may be helpful in reducing the use of additional ports.

There was no significant difference in the rates of hematoma formation, but hematomas were more frequently encountered in the LIHR group (n = 3) than the RIHR group (n = 1). For mesh fixation during inguinal hernia surgery, the mesh is usually fixed to the pubic bone or rectus muscle using a tacker. The assistant uses the tacker in robotic hernia surgery, and tacker misfires can cause bleeding and injury to surrounding organs. In our study, one tacker misfire occurred due to slipping during tacker fire, and we performed prolonged gauze compression to induce and confirm hemostasis. Given this concern, some surgeons prefer to use fibrin glue for mesh fixation or do not perform mesh fixation. However, it is necessary to pull the hernia sac and fix it to the pubic bone with a tacker to flatten the hernia sac in direct and other hernias. To prevent misfire, it is helpful to immobilize the assistant’s tacker with the opposite robot arm and guide positioning to prevent tacker slipping (Figure 3B).

Our study had some limitations, including the relatively small sample size (50 cases per group) and the fact that selection bias cannot be excluded in retrospective studies. Because robotic surgery is expensive, it was mainly chosen by people with personal health insurance. In South Korea, these people are relatively young and have a lot of interest in health. This selection bias occurred because a randomized control trial was impossible due to cost differences. However, our study was meaningful in that it was the first study on robotic inguinal hernia surgery conducted in South Korea. Recently, interest in robotic inguinal hernia surgery has increased in South Korea. Surgeons at various hospitals are introducing robotic inguinal hernia surgery into their practices, and a large number of multicenter studies on the effectiveness and safety of robotic inguinal hernia surgery are planned.

The authors should discuss the results and how they can be interpreted from the perspective of previous studies and the working hypotheses. The findings and their implications should be discussed in the broadest possible context. Future research directions may also be highlighted.

## 5. Conclusions

Robotic-assisted TAPP inguinal hernia surgery is a safe and efficient minimally invasive surgical procedure associated with a short learning curve. It can be learned without difficulty by surgeons who are proficient at laparoscopic inguinal hernia surgery. Also, robot inguinal hernia surgery is acceptable as a bridge operation for other, more complex robot surgeries.

## Figures and Tables

**Figure 1 medicina-59-00582-f001:**
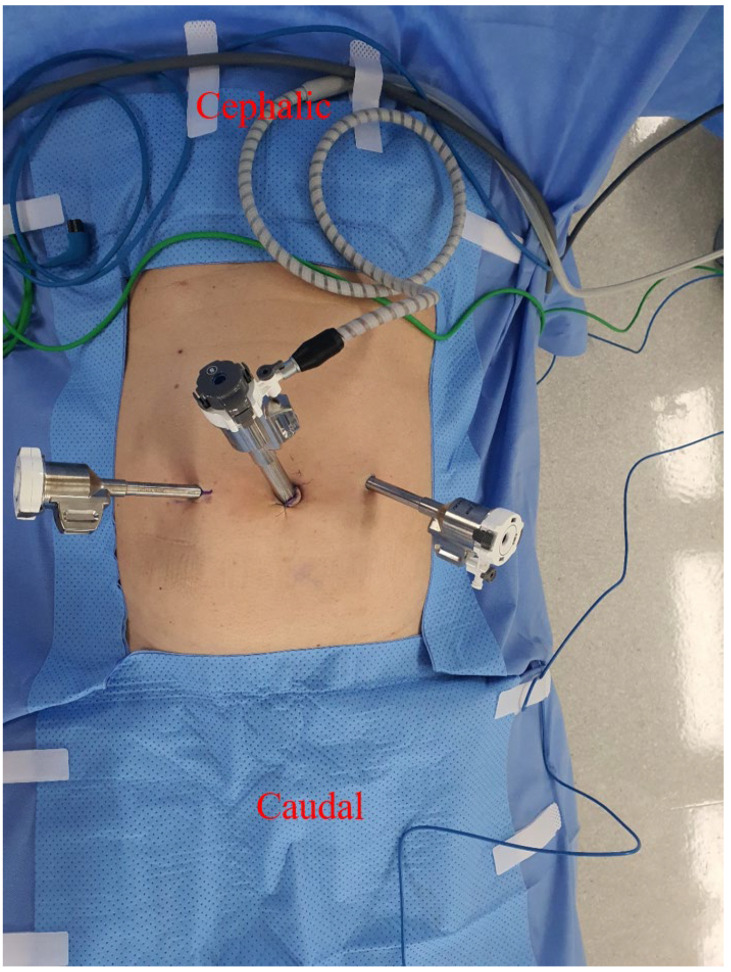
Robotic-assisted TAPP inguinal hernia surgery trocar placement.

**Figure 2 medicina-59-00582-f002:**
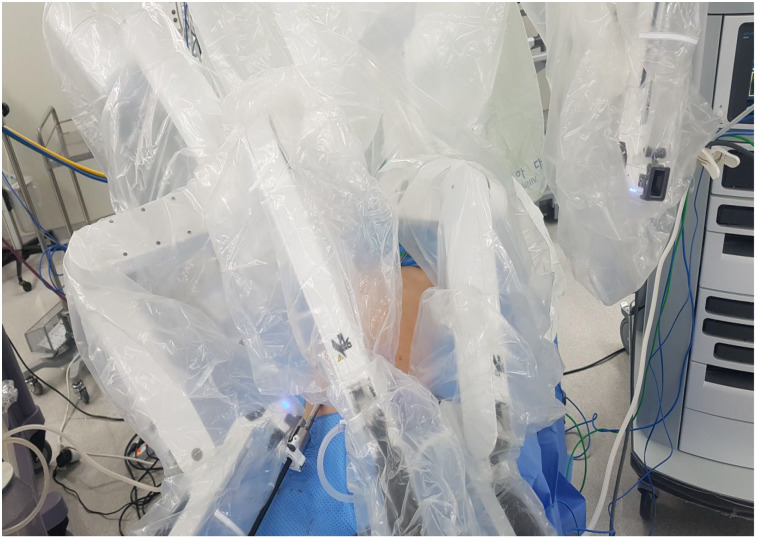
Robotic-assisted TAPP inguinal hernia surgery docking status.

**Figure 3 medicina-59-00582-f003:**
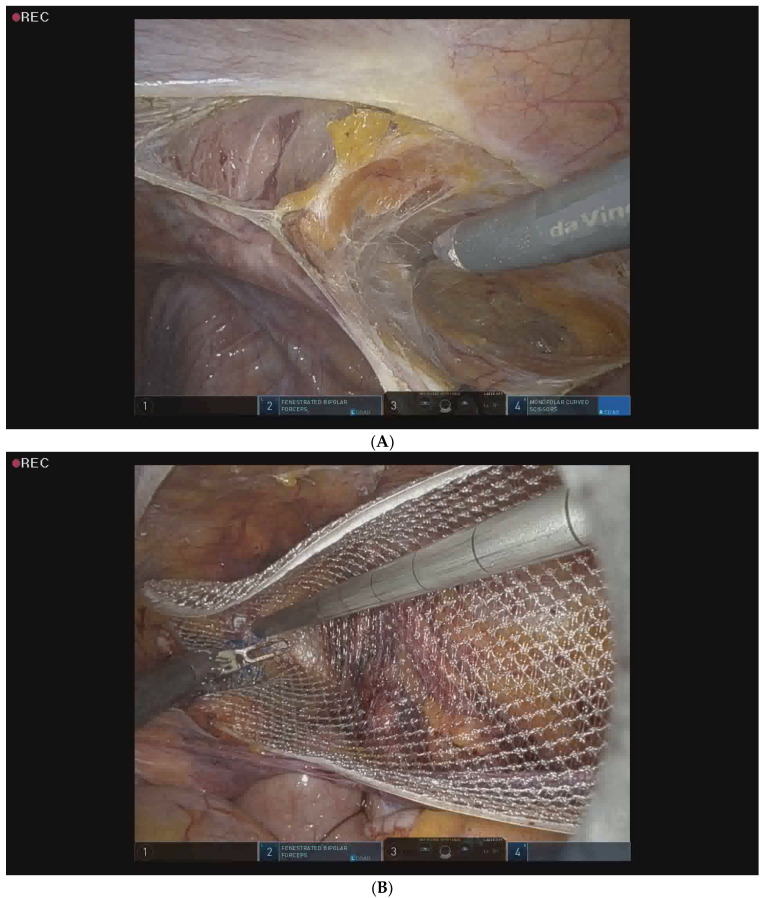
Robotic-assisted TAPP hernia repair procedure. (**A**). Peritoneum dissection using monopolar curved scissors. (**B**). Mesh application using a tack on the pubic bone. (**C**). Hernia sac fixation using a tack on the pubic bone. (**D**). Reperitonization using a robotic arm.

**Figure 4 medicina-59-00582-f004:**
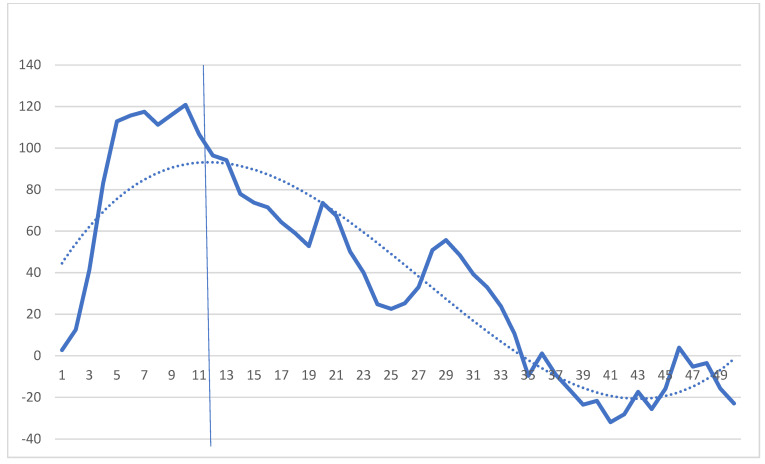
Cumulative sum (CUSUM) analysis of console times for robotic-assisted TAPP inguinal hernia surgery. The *x*-axis indicates consecutive cases, and the *y*-axis indicates the CUSUM score for robot console times. The vertical line represents the inflection point that divides between the early and late phases.

**Table 1 medicina-59-00582-t001:** Clinical characteristics of hernia patients.

Variables	All (n = 100)	Laparoscopic TAPP (n = 50)	Robotic TAPP (n = 50)	*p*-Value
Age (years, mean ± SD)	59.4 ± 15.2	64.4 ± 14.8	54.4 ± 14.0	0.001
BMI ^a^	24.3 ± 2.9	23.8 ± 2.9	24.8 ± 3.0	0.116
Gender (%)				
Male	99 (99%)	49 (98%)	50 (100%)	0.315
Female	1 (1%)	1 (2%)	0 (0%)	
ASA ^b^ score	2.2 ± 0.5	2.3 ± 0.6	2.0 ± 0.4	0.002
CLASS I (%)	5 (5%)	2 (2%)	3 (6%)	0.003
CLASS II (%)	73 (60%)	30 (60%)	43 (86%)	
CLASS III (%)	22 (100%)	18 (36%)	4 (8%)	
Comorbidities (%)				
HTN ^c^	44 (43%)	30 (60%)	14 (28%)	0.001
DM ^d^	14 (13%)	9 (18%)	5 (10%)	0.249
Cardiovascular	15 (15%)	10 (20%)	5 (10%)	0.161
Pulmonary	10 (10%)	7 (14%)	3 (6%)	0.182
Renal	0 (0%)	0 (0%)	0 (0%)	1
Liver	6 (6%)	3 (6%)	3 (6%)	1
Cerebral	4 (4%)	3 (6%)	1 (2%)	0.307
Other cancer history	10 (10%)	6 (12%)	4 (8%)	0.505
BPH ^e^	20 (20%)	12 (24%)	8 (16%)	0.317
Smoking	34 (34%)	15 (30%)	19 (38%)	0.398
Alcohol	42 (42%)	20 (40%)	22 (44%)	0.685
Steroid use	0 (0%)	0 (0%)	0 (0%)	1

^a^ Body mass index; ^b^ American Society of Anesthesiology; ^c^ Hypertension; ^d^ Diabetes mellitus; ^e^ Benign prostate hyperplasia.

**Table 2 medicina-59-00582-t002:** Intraoperative details.

Variables	All (n = 100)	Laparoscopic TAPP (n = 50)	Robotic TAPP (n = 50)	*p*-Value
Operation time (minutes, mean ± SD)	30.8 ± 11.1	31.5 ± 10.3	30.2 ± 11.9	0.56
Conversion rate (%) (Open or Laparoscopic)	0 (0%)	0 (0%)	0 (0%)	1
Cost of surgery (USD ^a^)		209.6 ± 27.5	3814.8 ± 172.9	<0.001
Type of hernia (%)				0.829
Indirect only	87 (87%)	44 (88%)	43 (86%)	
Direct only	7 (7%)	3 (6%)	4 (8%)	
Combined direct and indirect	3 (3%)	1 (2%)	2 (4%)	
Femoral	1 (1%)	0 (0%)	1 (2%)	
Spigelian	1 (1%)	1 (2%)	0 (0%)	
Hernia site (%)				0.483
Right	61 (61%)	32 (64%)	29 (58%)	
Left	31 (31%)	13 (26%)	18 (36%)	
Bilateral	8 (8%)	5 (10%)	3 (6%)	
Previous contralateral hernia (%)	10 (10%)	4 (8%)	6 (12%)	0.505
Complex hernia (%)				
Recurrent hernia	8 (8%)	5 (10%)	3 (6%)	0.461
Incarceration	9 (9%)	4 (8%)	5 (10%)	0.727
Prostatectomy history	8 (8%)	6 (12%)	2 (4%)	0.14

^a^ United States dollar.

**Table 3 medicina-59-00582-t003:** Postoperative outcomes.

Variables	All (n = 100)	Laparoscopic TAPP (n = 50)	Robotic TAPP (n = 50)	*p*-Value
Hospital stay (days, mean ± SD)	3.4 ± 1.6	3.5 ± 2.2	3.4 ± 0.6	0.658
VAS ^a^ score				
Operation day (0–10, mean ± SD)	4.7 ± 1.0	4.6 ± 1.1	4.8 ± 0.9	0.243
Postoperative 1 day (0–10, mean ± SD)	2.7 ± 0.7	2.9 ± 0.5	2.5 ± 0.7	0.015
Readmission within 30 days (%)	0 (0%)	0 (0%)	0 (0%)	1
Hernia recurrence (%)	1 (1%)	1 (2%)	0 (0%)	0.315
Postoperative outcome (%)				
Infection (Surgical site, Mesh)	0 (0%)	0 (0%)	0 (0%)	1
Seroma	2 (2%)	2 (4%)	0 (0%)	0.153
Hematoma	4 (4%)	3 (6%)	1 (2%)	0.307
Prolonged ileus	0 (0%)	0 (0%)	0 (0%)	1
Bowel obstruction	0 (0%)	0 (0%)	0 (0%)	1
Bladder injury	0 (0%)	0 (0%)	1 (2%)	0.315
Urinary retention	8 (8%)	5 (10%)	3 (6%)	0.461

^a^ visual analog scale.

## Data Availability

No new data were created or analyzed in this study from medical records.

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
