# Peer review of "Initial Experience of Robot-Assisted Transabdominal Preperitoneal (TAPP) Inguinal Hernia Repair by a Single Surgeon in South Korea"

_medicina, 2023, doi:10.3390/medicina59030582_

Round 1
Reviewer 1 Report
I must ask the authors to first modify the title of their manuscript by adding “hybrid laparoscopic” robotic-assisted transabdominal preperitoneal (TAPP) because that’s the technique they are describing, and this must be clearly differentiated from a fully robotic TAPP. Without this revision, I am afraid I will not be able to carry out further revision.
-Paragraph 2.5. and Figure 1. I have the impression the authors used a 12 mm robotic trocar for the camera, and I suppose for inserting the mesh inside the peritoneal cavity. If my impression is current, they must specify the different size of this trocar.
-page 5, line 138: the authors anchored the mesh to the pubic bone; I -personally- don’t have a problem with it as I used this technique as well. However, pros and cons must be discussed. For instance, the surgeons who do not opt for it claim a higher risk of osteitis associated with this strategy.
-page 5, line 139: the authors used a laparoscopic instrument to perform a step that could have been and should have been performed robotically as they anchored the mesh to the pubicle with a tacker. Beside the dangers associated with this approach, it significantly increases the costs of the procedure and defeats the whole purpose of performing the procedure robotically. It is for this reason that the technique used by the authors cannot and should be called fully robotic but rather hybrid, laparoscopic robotic-assisted”.
-page 8, line 175-176: the authors report that the mean cost of surgery was significantly higher in the robotic group. However, the extra cost related to the use of the laparoscopic tacker is not considered nor accounted for. The authors must provide this information and recalculate the cost accordingly (with and without tacker).
-page 11, lines 250-262: the authors should discuss cons of placing the camera port through the umbilicus vs. a para-umbilical position that for instance enables the surgeon to have a view more centered in respect to the target anatomy and could potentially eliminate the need for an additional laparoscopic assist trocar in an obese patient with a thick pannus and pre-peritoneal fat layer over the median umbilical ligament and fold.
Author Response
Answer>
Thank you very much for your review.
Your comment helped this paper a lot. In particular, the last comment will be reflected in our future surgery. Once again, thank you very much.
-I must ask the authors to first modify the title of their manuscript by adding “hybrid laparoscopic” robotic-assisted transabdominal preperitoneal (TAPP) because that’s the technique they are describing, and this must be clearly differentiated from a fully robotic TAPP. Without this revision, I am afraid I will not be able to carry out further revision.
We sincerely appreciate your comments.
We modified robot TAPP to robotic-assisted TAPP according to your comment.
-Paragraph 2.5. and Figure 1. I have the impression the authors used a 12 mm robotic trocar for the camera, and I suppose for inserting the mesh inside the peritoneal cavity. If my impression is current, they must specify the different size of this trocar.
We sincerely appreciate your comments.
Your comment about using a 12mm trocar is right. According to your comment, we have added the following to the text. Added information is indicated in blue text.
before
All operations were performed via the TAPP approach. Under general anesthesia, patients were placed in the supine position. Three trocars were inserted (Figure 1). The trocar locations were as follows: a camera port was inserted through the umbilicus, and 2 operating ports were inserted at both lateral aspects of the rectus abdominis muscle. The same locations were used for the robotic and laparoscopic groups. A distance of about 10 cm was maintained between the trocars.
after
All operations were performed via the TAPP approach. Under general anesthesia, patients were placed in the supine position. Three trocars were inserted (Figure 1). The trocar locations were as follows: a camera port was inserted through the umbilicus, and 2 operating ports were inserted at both lateral aspects of the rectus abdominis muscle. A 12mm trocar was used to insert mesh or gauze through the umbilicus. We tried to hide 12mm trocar wound using umbilicus. The same locations were used for the robotic and laparoscopic groups. A distance of about 10 cm was maintained between the trocars.
-page 5, line 138: the authors anchored the mesh to the pubic bone; I -personally- don’t have a problem with it as I used this technique as well. However, pros and cons must be discussed. For instance, the surgeons who do not opt for it claim a higher risk of osteitis associated with this strategy.
We sincerely appreciate your comments.
According to your opinion, we use a tacker on the pubic bone to avoid nerve injury caused by the tacker and fix the mesh firmly. Although we have no experience, we think your comment about ostitis is appropriate.
According to your comment, we have added the following to the text. Added information is indicated in blue text.
Before
After peritoneal dissection, we identified the hernia sac and dissected the hernia sac from the vas deference and testicular vessels. A large size mesh was applied to a large enough area to include the entire hernia site, and we fixed the mesh onto the pubic bone using a tacker. To prevent injury by tacker slippage during fixation, we immobilized the tacker with a robot arm before firing the tacker (Figure 3-B).
After
After peritoneal dissection, we identified the hernia sac and dissected the hernia sac from the vas deference and testicular vessels. A large size mesh was applied to a large enough area to include the entire hernia site, and we fixed the mesh onto the pubic bone using a tacker. We use a tacker on the pubic bone to avoid nerve injury caused by the tacker and fix the mesh firmly. Although we have no experience with this procedure, it should be considered about possibility of ostitis after surgery. To prevent injury by tacker slippage during fixation, we immobilized the tacker with a robot arm before firing the tacker (Figure 3-B).
-page 5, line 139: the authors used a laparoscopic instrument to perform a step that could have been and should have been performed robotically as they anchored the mesh to the pubicle with a tacker. Beside the dangers associated with this approach, it significantly increases the costs of the procedure and defeats the whole purpose of performing the procedure robotically. It is for this reason that the technique used by the authors cannot and should be called fully robotic but rather hybrid, laparoscopic robotic-assisted”.
We sincerely appreciate your comments.
We modified robot TAPP to robotic-assisted TAPP according to your comment.
-page 8, line 175-176: the authors report that the mean cost of surgery was significantly higher in the robotic group. However, the extra cost related to the use of the laparoscopic tacker is not considered nor accounted for. The authors must provide this information and recalculate the cost accordingly (with and without tacker).
Sincerely kinds you
In our hospital, the cost of robot surgery is fixed for each type of surgery. This cost includes all materials that were used during surgery. Therefore, there is no extra cost due to the use of the tacker you mentioned.
Please understand this point.
-page 11, lines 250-262: the authors should discuss cons of placing the camera port through the umbilicus vs. a para-umbilical position that for instance enables the surgeon to have a view more centered in respect to the target anatomy and could potentially eliminate the need for an additional laparoscopic assist trocar in an obese patient with a thick pannus and pre-peritoneal fat layer over the median umbilical ligament and fold.
We sincerely appreciate your comments.
Your valuable comments are something we didn't even think of. I will try to follow your advice in the future when performing surgery on obese patients.
Based on your comments, we have added the following to the text. Added information is indicated in blue text.
Before
The central camera port was inserted transumbilically to minimize scarring, and the 2 ports for the remaining robotic arm were inserted into the lateral aspects of the rectus abdominis muscle. Using these port locations, it is easy to operate on incidentally discovered contralateral hernias. In addition, surgeries such as cholecystectomy can be performed only by changing the direction of the robot docking without changing the port site. We performed robot-assisted TAPP inguinal repair on 2 patients with morbid obesity (BMI ≥35). In such cases, it is difficult to secure the operative field because of the severe visceral obesity, an additional assist port was used. The Intuitive guideline recommended that assist port is inserted at the level of the epigastric area between the 2 robot arms. An assistant must be placed between the robot arms in this situation, which can lead to frequent extracorporeal fighting between the robot arm and the assistant. In our experience, insertion of the assistant on the lateral side of the arm opposite the hernia site reduces this extracorporeal fighting.
After
The central camera port was inserted transumbilically to minimize scarring, and the 2 ports for the remaining robotic arm were inserted into the lateral aspects of the rectus abdominis muscle. Using these port locations, it is easy to operate on incidentally discovered contralateral hernias. In addition, surgeries such as cholecystectomy can be performed only by changing the direction of the robot docking without changing the port site. We performed robot-assisted TAPP inguinal repair on 2 patients with morbid obesity (BMI ≥35). In such cases, it is difficult to secure the operative field because of the severe visceral obesity, an additional assist port was used. The Intuitive guideline recommended that assist port is inserted at the level of the epigastric area between the 2 robot arms. An assistant must be placed between the robot arms in this situation, which can lead to frequent extracorporeal fighting between the robot arm and the assistant. In our experience, insertion of the assistant on the lateral side of the arm opposite the hernia site reduces this extracorporeal fighting. The para-umbilical camera port is considered to move toward the hernia site in morbid obesity patients. This can help to centralize the target anatomy and to avoid a thick pannus and preperitoneal fat layer over the median umbilical ligament in obese patients. This may be helpful reducing the use of additional ports.

Reviewer 2 Report
This is a well done report. I would, respectfully recommend a review of the prose in this report. There are some minor English language spelling and sentence structure corrections required.
Author Response
Thank you very much for your review.
We will do our best to do better research in the future.
We will work harder to perform more advanced robotic surgery.
Sincerely kind regards

Round 2
Reviewer 1 Report
I thank the authors for having addressed my kind inquiries. As result, I do believe that quality of their work has improved. However, an additional revision will be required.
- I reiterate the authors must modify the title of their manuscript by adding “hybrid laparoscopic robotic-assisted" transabdominal preperitoneal (TAPP) because that’s the technique they are describing, and this must be clearly differentiated from a fully robotic TAPP. The authors used a laparoscopic instrument to perform a step that could have been and should have been performed robotically as they anchored the mesh to the pubicle with a tacker. Beside the dangers associated with this approach, it significantly increases the costs of the procedure and defeats the whole purpose of performing the procedure robotically. It is for this reason that (I say this one more time) the technique used by the authors cannot and should not be called fully robotic but rather hybrid, laparoscopic robotic-assisted”.
-On page 8, line 175-176: the authors report that the mean cost of surgery was significantly higher in the robotic group. However, the extra cost related to the use of the laparoscopic tacker is not considered nor accounted for. The authors must provide this information and recalculate the cost accordingly (with and without tacker).
The consideration about internal allocation of resources at the authors’ medical center does not satisfy the reviewer's inquiry and is not appropriate for publication on a peer-reviewed and reputable scientific journal.
Regardless of how much the hospital covers for a specific procedure, that given procedure is associated by fixed and variable costs that have a numerical value. A fully robotic technique for inguinal hernia repair entails the use of one, maybe two non-absorbable sutures that are enormously cheaper than a laparoscopic tack applier. Thus, the additional cost related to the implementation of the presented hybrid technique through the use of a tack applier must be accounted for.
Author Response
I reiterate the authors must modify the title of their manuscript by adding “hybrid laparoscopic robotic-assisted" transabdominal preperitoneal (TAPP) because that’s the technique they are describing, and this must be clearly differentiated from a fully robotic TAPP. The authors used a laparoscopic instrument to perform a step that could have been and should have been performed robotically as they anchored the mesh to the pubicle with a tacker. Beside the dangers associated with this approach, it significantly increases the costs of the procedure and defeats the whole purpose of performing the procedure robotically. It is for this reason that (I say this one more time) the technique used by the authors cannot and should not be called fully robotic but rather hybrid, laparoscopic robotic-assisted”.
Answer>
We revised our title as your recommend.
Initial experience of hybrid laparoscopic robot-assisted transabdominal preperitoneal (TAPP) inguinal hernia repair by a single surgeon in South Korea
Sincerely thanks for your kind comment.
-On page 8, line 175-176: the authors report that the mean cost of surgery was significantly higher in the robotic group. However, the extra cost related to the use of the laparoscopic tacker is not considered nor accounted for. The authors must provide this information and recalculate the cost accordingly (with and without tacker).
Answer>
Thank you for your kind review.
The price of laparoscopic tacker is currently $400 in South Korea.
According to your comment, we have added the following to the text.
Added contents is highlighted in blue text.
Sincere thanks for your review
Before
Table 2 summarizes the operative details of the 2 groups. There was no significant difference in mean operation time between the LIHR and RIHR groups (31.52±10.31 ver-sus 30.22±11.87 min; p= 0.56). The mean cost of surgery was significantly higher in the RIHR group (209.61±27.52 US dollars versus 3814.75±172.97 US dollars; p< 0.001). All operations were completed according to the existing planned surgical method without open or laparoscopic conversion. There was no significant difference in hernia type be-tween the 2 groups. Indirect hernias were the most common, followed by direct hernias. There were 2 cases of combined hernias in the RIHR group and 1 case in the LIHR group. There was 1 case of femoral hernia in the RIHR group and 1 case of spigelian hernia in the LIHR group. Right-sided inguinal hernias were more common than left-side inguinal hernias in both groups. There were 5 cases of bilateral inguinal hernia in the LIHR group and 3 cases in the RIHR groups, respectively.
After
Table 2 summarizes the operative details of the 2 groups. There was no significant difference in mean operation time between the LIHR and RIHR groups (31.52±10.31 ver-sus 30.22±11.87 min; p= 0.56). The mean cost of surgery was significantly higher in the RIHR group (209.61±27.52 US dollars versus 3814.75±172.97 US dollars; p< 0.001). The RIHR group's cost of surgery includes the cost of several consumables and includes the price of a $400 laparoscopic tacker. All operations were completed according to the existing planned surgical method without open or laparoscopic conversion. There was no significant difference in hernia type be-tween the 2 groups. Indirect hernias were the most common, followed by direct hernias. There were 2 cases of combined hernias in the RIHR group and 1 case in the LIHR group. There was 1 case of femoral hernia in the RIHR group and 1 case of spigelian hernia in the LIHR group. Right-sided inguinal hernias were more common than left-side inguinal hernias in both groups. There were 5 cases of bilateral inguinal hernia in the LIHR group and 3 cases in the RIHR groups, respectively.
